# Radium Isotopes and Hydrochemical Signatures of Surface Water-Groundwater Interaction in the Salt-Wedge Razdolnaya River Estuary (Sea of Japan) in the Ice-Covered Period



Pavel Semkin *[ID], Pavel Tishchenko *, Alexander Charkin, Galina Pavlova, Yuri Barabanshchikov, Andrey Leusov, Petr Tishchenko [ID], Elena Shkirnikova and Maria Shvetsova

Il'ichev Pacific Oceanological Institute, Far Eastern Branch, Russian Academy of Sciences, Vladivostok 690041, Russia; charkin@poi.dvo.ru (A.C.); pavlova@poi.dvo.ru (G.P.); biw90@mail.ru (Y.B.); leusov.andrei@mail.ru (A.L.); eq15@poi.dvo.ru (P.T.); elmi@poi.dvo.ru (E.S.); chippers@rambler.ru (M.S.)

* Correspondence: pahno@list.ru (P.S.); tpavel@poi.dvo.ru (P.T.); Tel.: +7-(914)-664-7833 (P.S.)

**Abstract:** The interaction of surface water and groundwater is important in the ecology of coastal basins, affecting hydrological conditions, oxygen regime, carbon, and nutrient exchange. This study demonstrates a dynamic connection between the salt-wedge region and its underlying aquifer in the eutrophic estuary. In winter, this estuary is covered with ice, and the river flow is at its lowest; that is why the specific response to groundwater discharge is best marked in this season. Groundwater admixture was detected in the salt-wedge region by highly active radium isotopes: $^{223}$Ra—4.80 ± 0.42 dpm 100 L$^{-1}$, $^{224}$Ra—55.37 ± 1.1 dpm 100 L$^{-1}$, and $^{228}$Ra—189.71 ± 4.66 dpm 100 L$^{-1}$. The temperature of groundwater and river water was about +4 °C and 0 °C, respectively; that of seawater was −1.6 °C, and temperature increased up to +2.3 °C in the surface water–groundwater interaction region. Groundwater admixture is accompanied by a lower level of oxygen concentration of 52 μmol/kg; at that time, the maximum oxygen concentration in the salt-wedge region was 567 μmol/kg. In waters with a high activity of radium isotopes, there was a maximum partial pressure of $CO_2$—4454 μatm at the range 100–150 μatm in the salt-wedge region and also observed extremum of $NH_4^+$, $NO_2^-$, and dissolved phosphorus. The surface water–groundwater interaction through anoxic sediment can form localized anaerobic areas despite the general oxygen supersaturation of eutrophic estuary waters and also cause local recycling of nutrients from bottom sediments.

**Keywords:** salt-wedge estuary; surface water–groundwater interaction; radium isotopes; nutrients; $O_2$; $CO_2$



## 1. Introduction

Surface water–groundwater (SW-GW) interaction significantly influences the biogeochemistry of rivers [1], lakes [2], and coastal salt marshes [3], and the ecology along sea coasts around the world [4]. In coastal basins, SW-GW interaction has an effect on estuary ecosystems [5,6], including the formation of water with low dissolved oxygen (DO) in salt-wedge estuaries [7]. SW-GW interaction is often accompanied by temperature anomalies [8], and thus it can affect the aquatic biota [9,10].

Traditionally known SGD (submarine groundwater discharge), which implies SW-GW interaction in the subterranean estuary [11,12], may enhance benthic flux nutrients and dissolved organic carbon (DOC) [13,14]. Anoxic groundwater inputs account for oxygen deficit in bottom waters, and radium isotopes indicate that hypoxia is caused by offshore groundwater discharge [15–17]. The discharge of fresh groundwater in the global water balance is only 4.5 ± 3.2% from the discharge of river water to the ocean [18], but recirculated groundwater contributes 90% of the total water flow SGD [19].

Radium is an alkaline earth element that is adsorbed in river water, but exists as dissolved Ra$^{2+}$ in seawater due to ion exchanges [20]. Desorption of Ra from river suspen-

sion occurs at an early stage of the mixing zone of river and sea waters [21]. Therefore, the dissolved isotopes [224]Ra, [223]Ra, and [228]Ra, with a half-life of 3.6 days, 11.5 days, and 5.7 years, respectively, show a non-conservative dependence on water salinity in estuaries in the presence of river suspension [22]. The second source of dissolved radium for the estuary is SGD. Sea waters influenced by the Ra source in the diurnal and synoptic time ranges are enriched only in short-lived [224]Ra and [223]Ra isotopes, while the accumulation of long-lived [228]Ra isotopes takes years [22]. This makes it possible to establish the time scales of water dynamics and biogeochemical processes in coastal marine areas under the influence of SW-GW interaction and river runoff.

SW-GW interaction has been studied in lagoon estuaries [23–26] and in wetlands [27]. However, the study of this effect using radioisotopes in ice-covered basins is limited to only a few publications on the Arctic region [28,29]. The ice-covered estuarine ecosystems with a regime known as salt-wedge intrusion [30] have not been studied in terms of the impact of SW-GW interaction. In winter, the estuaries of the Japanese islands are subject to high water conditions [31], while the estuaries of the Russian coast are in low water and freeze-up conditions [32].

The Razdolnaya River Estuary is very eutrophic, and near-bottom hypoxia is observed during both low and high-water periods in summer [33,34]. However, during the freeze-up period, phytoplankton blooms occur with a chlorophyll "a" concentration of more than 100 µg/L, and then the concentration of DO is increased, and the partial pressure of $CO_2$ ($pCO_2$) is decreased [35]. The purpose of this article is to assess the impact produced by SW-GW interaction on the hydrological and hydrochemical conditions of the Razdolnaya River Estuary during the winter low water period covered by ice, based on radionuclides [223]Ra, [224]Ra, and [228]Ra. Thus, the novelty of this study is reflected in the following aspects: we study the SW-GW interaction in the area of the salt-wedge of the estuary with a long period of low water and ice cover; this estuary is highly eutrophic and phytoplankton blooms and superoxia/hypoxia occur near the bottom; we combine geochemical tracers such as three Ra isotopes and classical hydrological profiling performed with discrete synaptic variability during the coldest period of the year and the greatest ice thickness in the area when the groundwater signature is clear.

## 2. Materials and Methods

### 2.1. Study Area

The transboundary Razdolnaya River (China–Primorsky Territory of the Russian Federation) flows into the northern part of Amursky Bay (Peter the Great Bay, Sea of Japan) (Figure 1). The estuary measures about 50 km in length and is located within the boggy Razdolnenskaya depression and the northern part of Amursky Bay (Figure 1). The Razdolnaya River catchment area is 16,800 km$^2$. The average discharge of the river for the last 11 years has been 103.5 m$^3$/s (http://gmvo.skniivh.ru/ (accessed on 18 May 2022)). The water regime of Razdolnaya R. is characterized by steady low winter discharge of 7.2 and 5.7 m$^3$/s in January and February, respectively, and an absolute minimum discharge of 1.5 m$^3$/s in February. Discharge increases after the winter low water, usually in early March, but in the harsh winters from mid-March. Spring floods can be observed in May. The peaks of spring floods are about ten times higher than the average annual discharge of the river. The absolute discharge peaks exceed 3000 m$^3$/s during the summer and autumn floods in some years. In the course of freeze-up period from late November to early April, a salt wedge tends to penetrate the Razdolnaya River Estuary at a distance of up to 28–29 km from the river mouth bar [32,35]. During this period, the water salinity is more than 34 PSU in Amursky Bay and up to 26 PSU above the bar [32,35]. The average spring tide in Peter the Great Bay ranges from 15 to 20 cm, which makes it possible to classify the Razdolnaya River Estuary as a micro-tidal estuary with strong water stratification. The tides in the estuary of the Razdolnaya River are observed at a distance more them 20 km from the mouth river bar and were comparable with tides in Amursky Bay [36]. The currents are formed by tides in the estuary of the Razdolnaya River in the winter season. Change in the

direction of the currents was observed at the surface and at the bottom layer, depending on the tidal phase. During the flood tide, the current at the upper and bottom water layer at a velocity of up to 20 cm/s was oriented to the river; during the ebb-tide and low water level, direction of the current was opposite at a velocity of up to 20 cm/s [36]. At the same time, the daily variability of the penetration range of the saltwater wedge does not occur [36].

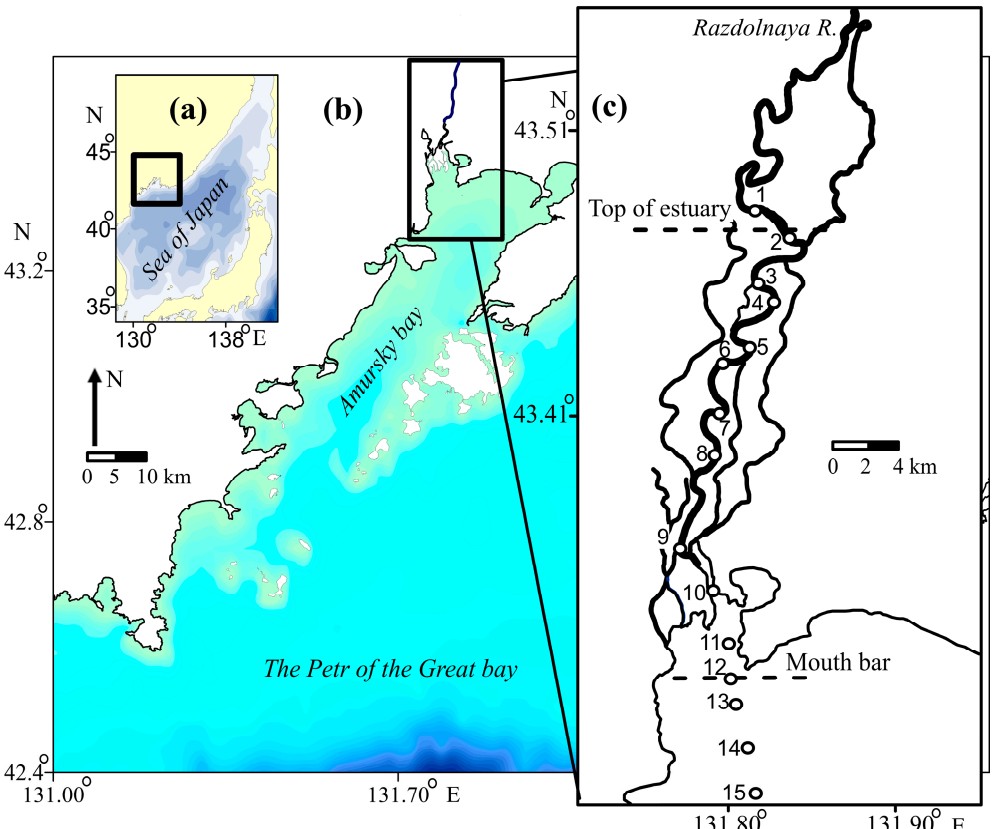

**Figure 1.** Map of study area: (**a**) Sea of Japan, (**b**) The Peter the Great Bay, and (**c**) position of monitoring stations at Razdolnaya R. estuary.

## 2.2. Field Work, Hydrological Surveys, and Water Sampling

We carry out a series of five surveys at 15 monitoring stations (St.) in Razdolnaya R. estuary (Figure 1) at about a week's intervals from 22 January 2022 to 23 February 2022 [35]. For each day, we estimated the state of the tide using data from the portal (http://esimo.oceanography.ru/tides/ (accessed on 18 May 2022)). To reduce the possible effect of diurnal variability in relation to tides, all data were taken in periods around high tide times. We probed the water with an RBR maestro multi-channel logger (RBR Ltd., Ottawa, ON, Canada) at an 8 Hz sampling rate. The following properties were logged: pressure, temperature, electrical conductivity, PAR, Chl-a fluorescence, chromophoric OM fluorescence, DO phosphorescence, and turbidity. The conversion from DO concentration to percent saturation was performed using the RBR software. On 22–23 February 2022, water samples were taken from the surface (under the ice) and bottom (0.2 m from the bottom) horizons using a 5-L Niskin bottle at 15 stations. Water samples were taken to measure radionuclides dissolved in water ($^{223}$Ra, $^{224}$Ra, and $^{228}$Ra), pH, total alkalinity (TA), nutrients (silica, phosphates, nitrates, nitrites, and ammonium, $P_{total}$ and $N_{total}$), salinity (S), and DOC.

We also took the water temperature in three water wells located in the estuary valley. Prior to measurements, we pumped out water from these wells to measure characteristics of the inflowing groundwater. The value of 4 °C was typical of groundwater temperatures.

We made an experiment on the desorption of radium isotopes from river suspension by adding salt to a sample of river water at St. 1 to salinity of 21.5 psu. The resulting sample was filtered one day after sampling and included in the total series of measurements of radium isotopes.

Four sediment cores were taken with a gravity tube at stations 2, 4, 5, and 10 for visual description and estimates of the activity of radium isotopes in sediments [37].

### 2.3. Radium Isotopes Measurements

Water samples with a volume of 44 L were filtered through 1 mm polypropylene cartridges and Mn fiber at a flow rate of maximum 1 L min$^{-1}$ to obtain a Ra extraction efficiency of at least 97% [38]. Radioisotopes $^{224}$Ra and $^{223}$Ra were measured using a Radium Delayed Coincidence Counter (RaDeCC, West Columbia, SC, USA) [39]. To make correction for the supported $^{224}$Ra, we conducted the second set of measurements in 2 to 6 weeks so that the initial excessive activity of $^{224}$Ra (ex $^{224}$Ra) could reach secular equilibrium with $^{228}$Th, which was also absorbed on Mn fiber [39]. The activity of $^{228}$Ra was measured after 5–6 months, with correction for the decay of $^{228}$Th, which had initially been sorbed onto Mn fiber from original samples [38]. To calibrate RaDeCC systems for $^{224}$Ra, $^{223}$Ra, and $^{228}$Ra, measurements using $^{232}$Th standards as described by [40] were used.

### 2.4. Analysis of Hydrochemical Characteristics

The $NH_4^+$ concentration was determined using the indophenols method. $NO_3^-$, $NO_2^-$, dissolved silicates (DSi), and dissolved inorganic phosphorus (DIP) were measured using standard colorimetric methods. Optical density was determined on a KFK-3KM photocolorimeter (Ruspribor, Saint Petersburg, Russia). Details of the methods used for the nutrient analyses are given in Grasshoff et al. [41]. The sum of the $NH_4^+$, $NO_3^-$, and $NO_2^-$ concentrations is the dissolved inorganic nitrogen (DIN). The detection limit was 0.01 μmol/L for the phosphate and DIN, and 0.02 μmol/L for DSi. The total phosphorus ($P_{total}$) and total nitrogen ($N_{total}$) were determined by persulfate combustion with ultraviolet irradiation on a SKALAR SAN++ analyzer (Skalar, Breda, The Netherlands). The detection limit was 0.01 μmol/L for the $P_{total}$ and $N_{total}$.

Water samples for determining DOC were prefiltered through a glass filter with pore size of 0.6 μm. The analysis used automated analyzer Shimadzu TOC-VCPN (Shimadzu, Kyoto, Japan). Salinity was also measured by salinometer Guildline Autosal 8400B (Guildline Instruments, Smiths Falls, ON, Canada) with accuracy of 0.002.

The $pCO_2$ was calculated from the measured pH and TA using a commonly known procedure [42]. A potentiometric method was applied to determine pH on InoLab pH/ION/Cond 750 m (Laborkomplekt, Moscow, Russia). pH was measured at 10 °C using a cell without a liquid junction [43]. The precision of pH measurements was about ±0.004 pH units. TA analysis was carried out through direct colorimetric titration on Dosimat-665 (Metrohm, Herisau, Switzerland) with hydrochloric acid in an open cell according to Bruevich's method [43,44]. TA measurements were performed with a precision of ±3 μmol/kg.

The software used for statistical analyses was MS Excel 2019. The spatial distribution maps were developed using the program Surfer 9 (Golden Software).

## 3. Results

### 3.1. Hydrological Conditions

The observations were made during a steady state salt-wedge penetration to about 29 km from the river mouth bar (Figure 2) [35]. Further, salt-wedge penetration into the estuary was limited to the sandy channel between St. 1 and St. 2, with a depth of less than 0.5 m. As our observations showed, it was frozen to the bottom. There was a general decrease in the salinity of the bottom layer, which was most pronounced for St. 10, where we observed a decrease from 30 to 25 PSU (Figure 2). Freshening of the bottom layer was accompanied by a gradient decrease in the halocline layer, which can be distinguished by the salinity jump from 8 to 25 PSU (Figure 2).

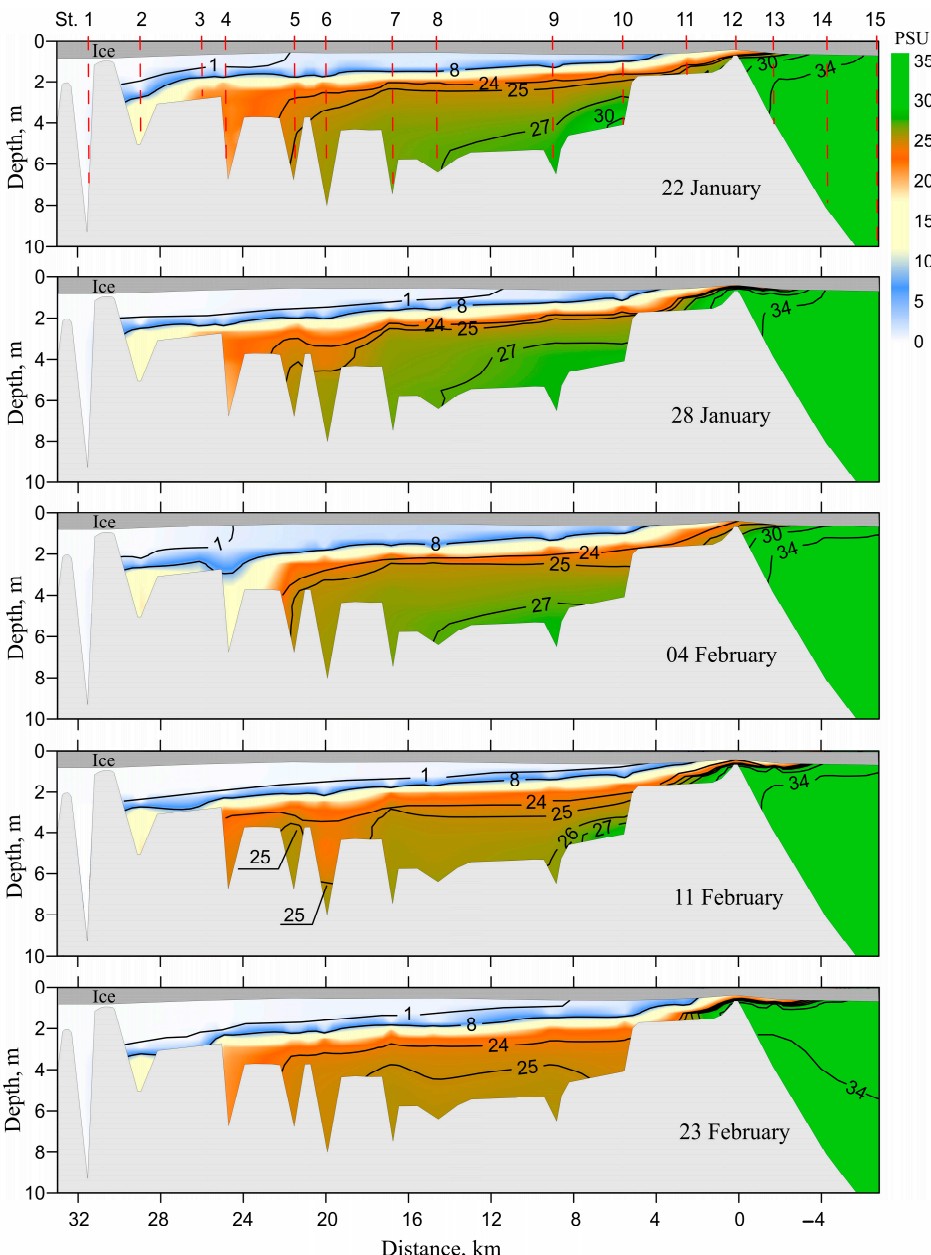

**Figure 2.** Salinity data on transect obtained from 22 January 2022 to 23 February 2022. The dates shown in the figure indicate the day of profiling. The position of monitoring stations is shown by dotted red lines in first graph. Station site locations are indicated in Figure 1. Positive values on the *X*-axis—distance from the river mouth bar (St. 12) to upstream; negative values—to downstream.

The temperature of sea and river waters was about −1.6 °C and 0 °C, respectively (Figure 3). The near-bottom layer temperature increased up to +2.3 °C at St. 2 during the observation period at the salt-wedge region [35].

The ice thickness on the transect as a whole increased during the observation period (Table 1) [35]. However, in the area of the bar on St. 12, the thickness of the ice remained almost unchanged or even decreased in early February. In this case, the water exchange rate between the estuary channel and the receiving basin decreases. The river discharge during the ice formation period is minimized, and the estuary, in this case, can be compared to a closed lake system to a certain extent.

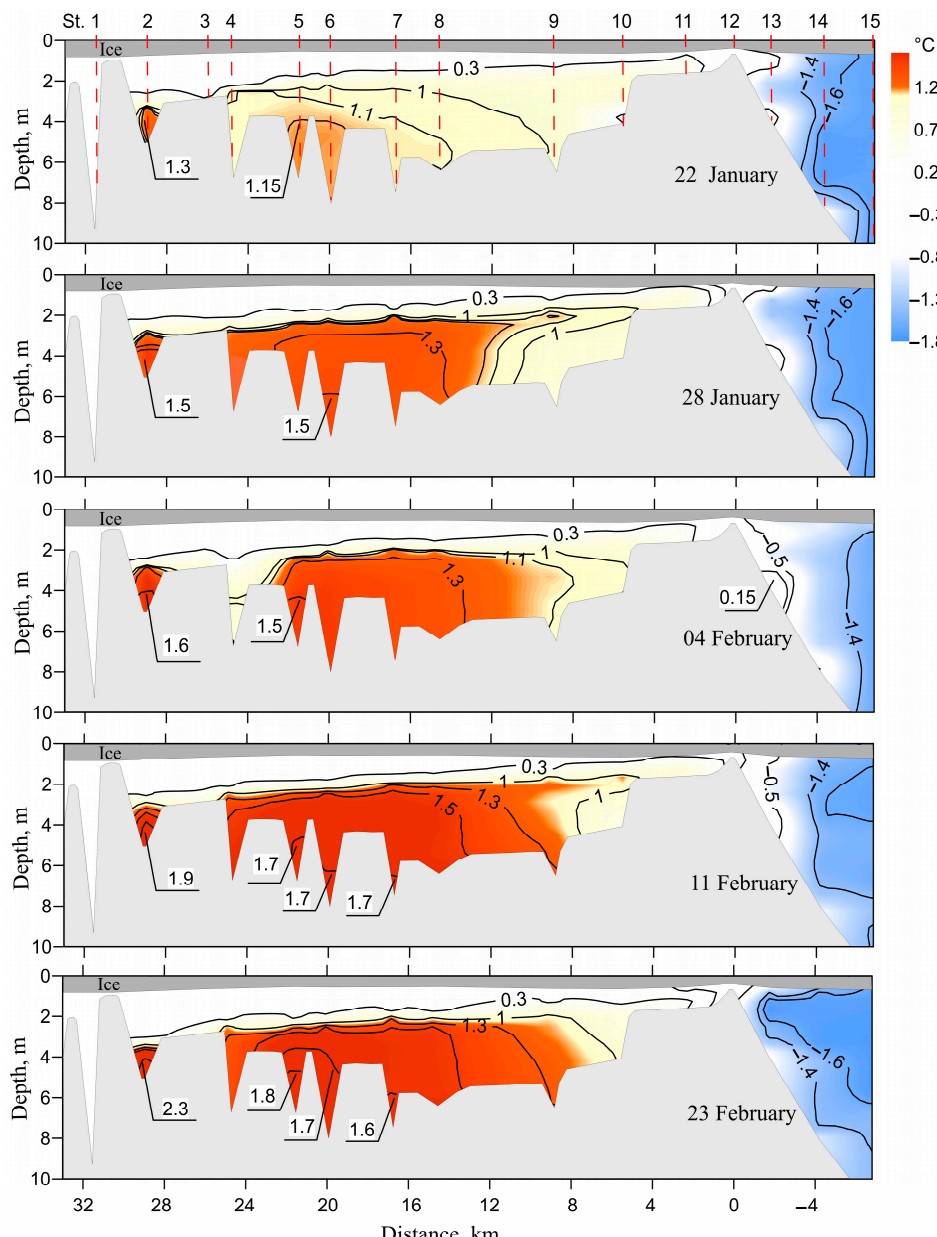

**Figure 3.** Water temperature on transect obtained from 22 January 2022 to 23 February 2022. The dates shown in the figure indicate the day of profiling. The position of monitoring stations is shown by dotted red lines in first graph. Station site locations are indicated in Figure 1. Positive values on the *X*-axis—distance from the river mouth bar (St. 12) to upstream; negative values—to downstream.

**Table 1.** Ice thickness (cm).

| | Station Numbers | | | | | | | | | | | | | | |
|---|---|---|---|---|---|---|---|---|---|---|---|---|---|---|---|
| **Dates** | **1** | **2** | **3** | **4** | **5** | **6** | **7** | **8** | **9** | **10** | **11** | **12** | **13** | **14** | **15** |
| 22 January | 75 | 70 | 65 | 60 | 65 | 64 | 56 | 56 | 60 | 56 | 50 | 40 | 53 | 48 | |
| 28 January | 79 | 73 | 66 | 64 | 65 | 65 | 58 | 56 | 61 | 58 | 51 | 42 | 56 | 51 | 45 |
| 4 February | 83 | 75 | 68 | 65 | 67 | 67 | 56 | 57 | 62 | 57 | 50 | 35 | 50 | 55 | 49 |
| 11 February | 81 | 76 | 70 | 69 | 70 | 70 | 56 | 55 | 62 | 58 | 50 | 40 | 64 | 60 | 54 |
| 23 February | 90 | 79 | 66 | 67 | 69 | 69 | 55 | 55 | 61 | 59 | 52 | 41 | 70 | 66 | 60 |

### 3.2. Radium Isotopes Activity and Distribution of Hydrochemical Characteristics

As seen in Table 2, the activity of short-lived isotopes $^{224}Ra$ and $^{223}Ra$ increased in the sample of river water to which we added salt; however, the activity of the long-lived isotope $^{228}Ra$ was unchanged. The highest total activity rate of radium isotopes $^{224}Ra$, $^{223}Ra$, and $^{228}Ra$ is observed at St. 2, and an increased activity rate is observed all across the topographic depression with its salt wedge at St. 2–10. Therefore, the highest activity of all three isotopes of radium on St. 2 is mainly associated with the influence of SW-GW interaction and, secondarily, with ion-exchange processes with an increase in salinity. The names of water types are presented based on the activity of isotopes of radium in endmember water (Table 2). Ratios close to equilibrium, $^{224}Ra/^{223}Ra$ = 21.7 and $^{224}Ra/^{228}Ra$ = 0.67, were obtained for St. 4, which indicates the admixture of recent groundwater in this region. The highest activity of $^{228}Ra$ was found at St. 2, with $^{224}Ra/^{228}Ra$ = 0.2. That is, in this place, $^{228}Ra$ accumulated without the addition of $^{224}Ra$. Generally, we obtained the irregular distribution of the three isotopes and their ratios, probably associated with the irregular inflow of groundwater in different places and the transport of these waters by currents up and down the channel depending on the tides (Table 2).

**Table 2.** Depth (m) of sampling, salinity (S) (PSU), and activities radium isotopes (dpm 100 $L^{-1}$). RW: river water, RW + Salt: river water sample with added salt, SWR: salt-wedge region, SW: seawater, and HTBW: high-turbidity brackish water (mixture of RW and SW). Date of sampling: 23 February 2022.

| St. №/Water Mass | Depth of Sampling | S | ex$^{223}Ra$ | ex$^{224}Ra$ | $^{224}Ra/^{223}Ra$ | $^{228}Ra$ | $^{224}Ra/^{228}Ra$ |
|---|---|---|---|---|---|---|---|
| 1/RW | 8.3 | 0.14 | 0.05 ± 0.009 | 7.89 ± 0.19 | 157.8 | 11.95 ± 0.82 | 0.66 |
| 1/RW + Salt | 8.3 | 21.5 | 4.38 ± 0.38 | 37.27 ± 1.04 | 8.5 | 7.84 ± 0.56 | 4.76 |
| 2/SWR | 5 | 15.23 | 4.80 ± 0.42 | 38.74 ± 1.09 | 8.1 | 189.71 ± 4.66 | 0.2 |
| 4/SWR | 6.8 | 21.09 | 2.55 ± 0.23 | 55.37 ± 1.1 | 21.7 | 82.29 ± 2.1 | 0.67 |
| 5/SWR | 6.3 | 24.19 | 1.36 ± 0.18 | 27.01 ± 0.82 | 19.9 | 79.62 ± 3.26 | 0.34 |
| 6/SWR | 7.8 | 24.56 | 0.70 ± 0.11 | 30.00 ± 0.87 | 42.9 | 49.81 ± 1.48 | 0.6 |
| 7/SWR | 7.3 | 24.99 | 0.79 ± 0.13 | 22.31 ± 0.87 | 28.2 | 81.95 ± 4.77 | 0.27 |
| 8/SWR | 6.7 | 25.03 | 0.33 ± 0.07 | 20.76 ± 0.84 | 62.9 | 61.57 ± 3.32 | 0.34 |
| 9/SWR | 6.8 | 25.13 | 0.79 ± 0.06 | 15.12 ± 0.24 | 19.1 | 68.70 ± 3.29 | 0.22 |
| 10/SWR | 4 | 22.88 | 1.39 ± 0.06 | 19.64 ± 0.29 | 14.1 | 55.44 ± 1.97 | 0.35 |
| 11/HTBW | 1.3 | 19.25 | 0.64 ± 0.04 | 16.57 ± 1.39 | 25.9 | 5.92 ± 0.42 | 2.8 |
| 12/HTBW | 0.5 | 19.1 | 0.41 ± 0.03 | 10.91 ± 0.67 | 26.6 | 21.11 ± 1.50 | 0.52 |
| 13/SW | 3.9 | 32.52 | 0.15 ± 0.015 | 2.37 ± 1.63 | 15.8 | 11.58 ± 0.92 | 0.2 |
| 14/SW | 8 | 34 | 0.13 ± 0.004 | 2.13 ± 0.03 | 16.4 | 20.44 ± 2.12 | 0.1 |
| 15/SW | 1 | 31.15 | 0.08 ± 0.015 | 1.72 ± 0.087 | 21.5 | 23.40 ± 1.63 | 0.07 |

A minimum oxygen saturation of about 21%, maximum $pCO_2$ of 4500 µatm, and maximum radium isotope activity exist in the estuary's near-bottom waters (Figure 4). It was a local anomaly in the St. 2 area. At the same time, water is oversaturated with oxygen, from 110 to 150%, in seawater and at stations in the region of the salt wedge (Figure 4) due to the phytoplankton bloom outbreak [35].

Figure 5 shows the non-conservative relationship of nutrients to salinity. $N_{total}$ and DIN markers at St. 2 and in several samples from the surface layer are located above the mixing line, which indicates their addition. The remaining markers are predominantly located below the mixing line, indicating the intake of $N_{total}$ in the mixing zone of river and sea waters. For $P_{total}$ and DIP, St. 2 is located below the mixing line and other markers.

The maximum concentrations of $NH_4^+$ (73.7 µmol/L) and $NO_2^-$ (2.1 µmol/L) were observed in waters with the highest activity of $^{228}Ra$ at St. 2 (Figure 6). Other nutrients and DOC also increase in St. 2 (Figure 6).

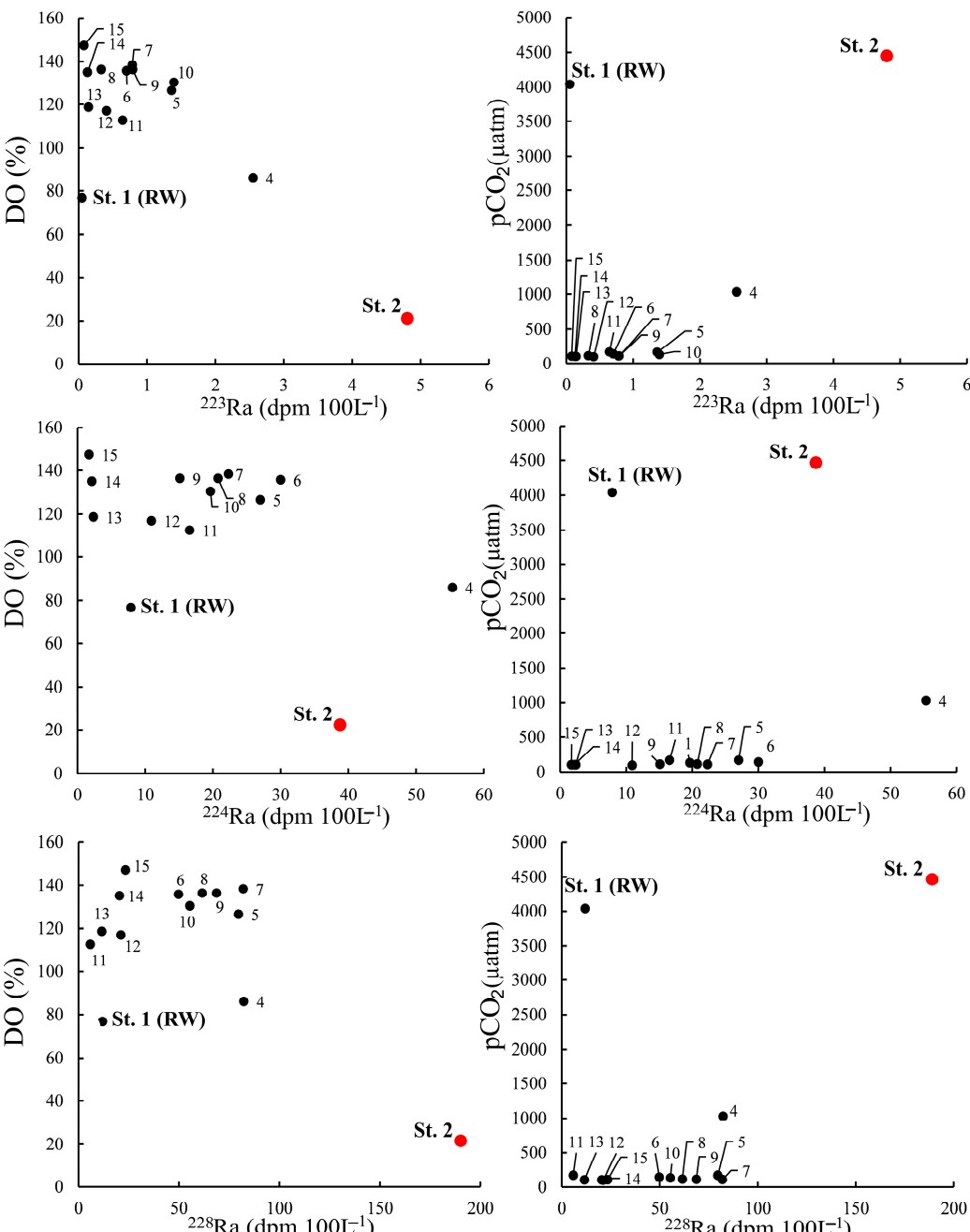

**Figure 4.** Dependence between DO (% saturation) and $pCO_2$ on the activity of $^{223}$Ra, $^{224}$Ra, and $^{228}$Ra. 23 February 2022.

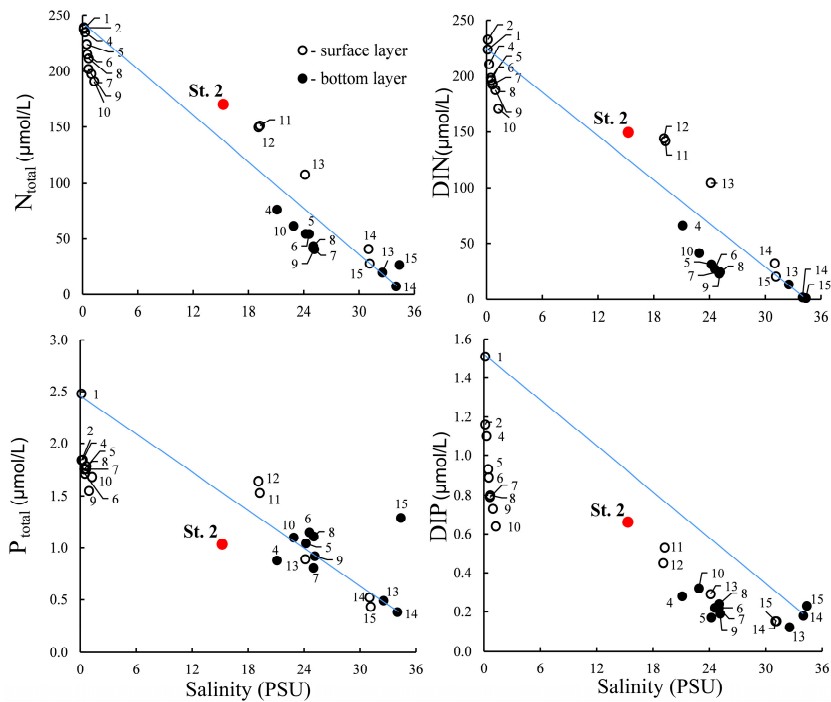

**Figure 5.** Dependence of nutrient concentrations on salinity. Blue lines connect markers for river and sea waters. 23 February 2022.

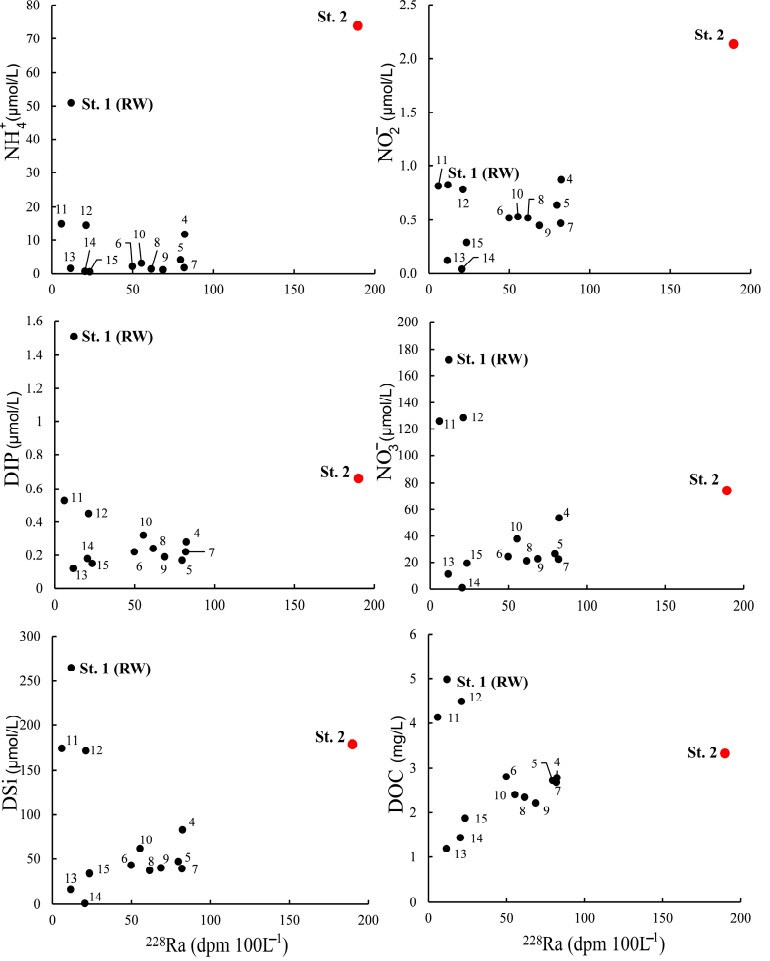

**Figure 6.** Dependence between nutrients and DOC on the activity of $^{228}$Ra. 23 February 2022.

## 4. Discussion

### 4.1. Identification and Control Factorsof SW-GW Interaction

The combination of high radium isotope activity rate (Table 2) and temperature anomaly (Figure 3) shows that the water of the salt wedge is formed as a result of the admixture of groundwater.

However, we accept the following assumptions. It has been reported that saline waters interact with sediments, which are a source of radium isotopes, through various processes such as erosion, diffusion, and bio irrigation for the coastal embayment [25]. Other studies confirm that the contribution of $^{228}$Ra due to erosion remains the main one for the shelf [45], including our results obtained for the summer flood in the Razdolnaya River Estuary [46]. The role of SW-GW interaction in the export of short-lived radium isotopes in the Razdolnaya River Estuary increases in some of the deepest areas, for example, the $^{224}$Ra maximum at St. 4 (Table 2). However, the contribution of pore exchange to the export of $^{224}$Ra for ichthyofauna exists in the Razdolnaya River Estuary [46]. The listed factors suggest careful use of $^{223}$Ra, $^{224}$Ra, and $^{228}$Ra as groundwater tracers if there are no additional evident indicators, such as water temperature in our case.

It is known that relatively short-period seawater recirculation in the upper aquifer is accompanied by enrichment with short-lived radium isotopes $^{224}$Ra and $^{223}$Ra. Admixing salt water in this aquifer will bring the long-lived isotope $^{228}$Ra, but long-term processes are essential for the significant accumulation of $^{228}$Ra [12,25,38,47]. The composition of stable isotopes $\delta^{18}$O and $\delta$D in the SW-GW interaction zone is subject to the seawater/river water ratio [48]. It is considered that the main reason for the increase in all three isotopes $^{223}$Ra, $^{224}$Ra, and $^{228}$Ra is that water recirculated into the upper aquifer during the winter period. The upper, highly permeable aquifer comprises pebbles and sand, and its thickness is 20 m [49]. This is the alluvial–marine complex that extends 30 km from the Razdolnaya River mouth bar, and seawaters penetrate into it in winter along the estuary. The low-water period and salt-wedge regime begin from September–October [33], and then the seawater penetrates for many kilometers from the bar, as seen in Figure 2. Salinization of the upper aquifer in the estuary area for 5 months from September–October to February leads to the accumulation of dissolved long-lived $^{288}$Ra isotopes in it. The subsequent recirculation of these waters will lead to an observed increase in all three isotopes with an increase in water temperature in winter (Figure 3). Salinity in the bottom layer of estuary water decreases during the low water period, since there is inflow in the estuary by fresh groundwater from the land side, but there is no inflow of cold salt water (Figures 2 and 3).

As a result of SW-GW interaction, we observed an increase in all three isotopes $^{223}$Ra, $^{224}$Ra, and $^{228}$Ra but in different intensities at various stations. We believe that the SW-GW interaction in the Estuary of the Razdolnaya River is in the areas with sand sediments, similar to other permeable sand systems [50]. Exactly at St. 2 and 4, we found sand and silty sand in the core, in contrast to other areas of the Razdolnaya River Estuary, where gray silt can be found. In addition, the sandbars along the estuary located between our stations throughout the salt wedge (Figure 2) are also completely composed of sand. We believe that SW-GW interaction with the underlying aquifer occurs in the local deepest section lines of the estuary and probably also on sandbars between our stations.

The circulation of groundwater has a compound system in each local region, while the recharge of the upper aquifer is usually determined by the intensity of rainfall everywhere [51]. The favorable conditions for salt-wedge water recirculation in the upper aquifer exist during the winter discharge low period [52] because the coastal aquifer along Razdolnaya River Estuary is the least loaded with meteoric water during this period [49]. It is known that tides generate complex groundwater fluctuations in aquifers [52]. In turn, semidiurnal variations in recirculated groundwater flux also have a period of tidal sea level fluctuations [12,47,53]. In the salt-wedge region of the Razdolnaya River Estuary, semidiurnal tides are observed [36]. If the salt groundwater meets with fresh land waters in the upper aquifer, density gradients in the transitional zone cause convective circulation, and groundwater discharge occurs [54,55].

Thus, the main drivers for SW-GW interaction increasing in the salt-wedge region of Razdolnaya River Estuary during the freeze-up period are as follows: increased proportion of salt water in the upper aquifer along of estuary since autumn, slow water dynamics as a result of ice-covered and low river discharge, increased duration of the hydraulic head during the period of high tide and low tide, and the high hydraulic conductivity of the underlying aquifer.

### 4.2. River Water, Seawater, and Groundwater Fractions in the Bottom Water Layer

The river water, seawater, and groundwater contributions can be quantified by applying a mass balance calculation [56] based on $^{228}$Ra and salinity data. It is assumed that each sample is a mixture of marine water (fSW), river runoff (fRW), and SG (fSG). The balance is governed by the following equations:

$$f_{SW} + f_{RW} + f_{SG} = 1$$

$$f_{SW} \times S_{SW} + f_{RW} \times S_{RW} + f_{SG} \times S_{SG} = S_{meas}$$

$$f_{SW} \times Ra_{SW} + f_{RW} \times Ra_{RW} + f_{SG} \times Ra_{SG} = Ra_{meas}$$

where $f_{SW}$, $f_{RW}$, and $f_{SG}$ are the fractions of marine water, river runoff, and SG in a water parcel, and $S_{SW}$, $S_{RW}$, $S_{SG}$, $Ra_{SW}$, $Ra_{RW}$, and $Ra_{SG}$ are the corresponding salinity and $^{228}$Ra activity. Smeas and Rameas are the measured values of salinity and $^{228}$Ra of the water samples. A special selection of endmember values is required for the Razdolnaya River Estuary in the ice-covered period (see Table 2). The $^{228}$Ra activity of sediments is taken 270 dpm 100 L$^{-1}$ in St. 2. We used this isotope also because the increase in its activity was not associated with suspended matter, as shown by the experiment on adding salt to river water (see Table 2).

Figure 7 shows the admixture of groundwater in the salt-wedge region, the fractions of which increase from the sea with decreased salinity (Figure 2). The fraction of groundwater prevails in St. 2 (Figure 7), where the water temperature was also the highest (Figure 3).

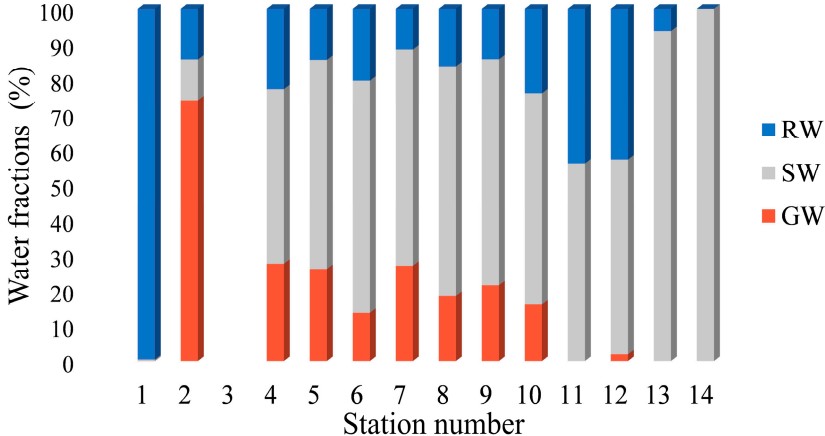

**Figure 7.** Distribution of water fractions in the bottom layer (RW—river water, SW—sea water, GW—groundwater). Station site locations are indicated in Figure 1. 23 February 2022.

### 4.3. Potential Estuarine Ecosystem Response to SW-GW Interaction

We suggest that the groundwater seepage through the anoxic sediments leads to a decrease in the DO of groundwater, similar to processes shown and discussed in other papers [4,15–17,37]. As a result, in the course of SW-GW interaction between the upper aquifer and salt-wedge water, we see DO saturation rates in the near-bottom layer reducing toward the estuary top, and a simultaneous increase in water temperatures and radium isotopes activity (Table 2, Figures 3 and 4). In the warm season, the salt wedge in the Razdolnaya

River Estuary has a DO close to zero due to the intensification of production-destruction processes as well as to the high turbidity of water and, accordingly, photosynthetically active radiation close to zero of the near-bottom water layer [33]. However, suspended matter concentrations are relatively low in winter, and the photic zone normally extends to the bottom everywhere, with nutrient concentrations being very high [32]. That is why photosynthesis prevails over organic matter destruction, and DO oversaturation is observed in the Razdolnaya River Estuary in winter [35]. Not excepting that the key factor of DO-specific conditions for the Razdolnaya R. estuary is the balance of organic matter production–destruction, results (Figure 4) are indicative of SW-GW interaction influence on the formation of DO-specific and high $pCO_2$ conditions of the salt-wedge region.

This occurs when DO is depleted in sediments, and the distraction of organic matter occurs in the following reactions: denitrification, reduction of irons (Fe) and manganese oxides, reduction of sulfates, or methanogenesis [57]. Estuary sediments Razdolnaya River Estuary are enriched with iron as a result of their flux with river water [58]. When the salt water penetration into sediments, the destruction of organic matter occurs with the reduction of sulfates and the accumulation of $NH_4^+$ [59]. The destruction of organic matter in the sediment leads to an increase in the concentration of DIP in the pore water.

In the winter period, the dissolved nitrogen and phosphorus concentrations of the Razdolnaya R. are up to 500 and 9 $\mu$mol/L, respectively, because the flux of nutrients from municipal wastewater exists [60]. Eutrophication of river waters is accompanied by the accumulation of nutrients and organic matter in the sediments of the estuary. Therefore, the concentration of $NH_4^+$ in the pore water of sediments reaches 2088 $\mu$mol/L, and the $pCO_2$ value is 11,739 $\mu$atm [61]. DIP in sediments is usually sorbed, but the redox potential strongly influences the dissolution of DIP in sediments [62]. In sediments from eutrophic basins, the redox boundary can be located within a few millimeters of the upper sediment layer, and DIP is released into the pore water below this boundary [63]. Thus, Fe compounds can be a source or sink of DIP in bottom sediments depending on the oxygen content there. This can explain the local DIP extremum with the extremums of $NH_4^+$ and $NO_2^-$ in the SW-GW interaction region at St. 2 (Figures 6 and 7).

## 5. Conclusions

The SW-GW interaction is estimated with a geotracer-based model, including a series of hydrological profilings conducted to estimate the synaptic variability of a complex of hydrological and hydrochemical characteristics.

This study highlights the fundamental role of the SW-GW interaction in the formation of the temperature and hydrochemical regime of an ice-covered estuary when the estuary water exchange is significantly limited, and mixing due to wind is excluded. SW-GW interaction through anoxic sediment can form localized anaerobic areas despite the general oxygen supersaturation of eutrophic estuary waters, and the following important conclusions were drawn:

1.  SW-GW interaction has a significant effect on the ecological state of the estuary because it is accompanied by a response in the concentration of DO, $CO_2$, and nutrients, thus forming local recycling from sediments;
2.  The SW-GW interaction can be an additional source of organic matter from sediments;
3.  SW-GW interaction increases the temperature of the salt-wedge region in the ice-covered estuary.

We also obtained various ratios of $^{224}Ra/^{223}Ra$ and $^{224}Ra/^{228}Ra$ throughout the estuary. High activity of $^{224}Ra$ and $^{223}Ra$ in some cases was associated with desorption from suspension. The high activity of isotope $^{228}Ra$, together with temperature anomalies, most clearly indicated the admixture of groundwater in the salt wedge area. The results indicate a variety of SW-GW interactions during the freezing period, probably due to the tidal dynamics of waters and depth variability throughout the estuary.

**Author Contributions:** P.S.: writing—original draft, conceptualization, methodology, formal analysis, investigation, and funding acquisition. P.T. (Pavel Tishchenko): writing—review and editing, investigation, conceptualization, supervision, funding acquisition. A.C.: writing—review and editing, investigation, formal analysis, funding acquisition, and methodology. G.P.: writing—review and editing, investigation, and formal analysis. Y.B.: methodology, formal analysis, and investigation. A.L.: methodology, formal analysis, and investigation. P.T. (Petr Tishchenko): writing—review and editing, and investigation. E.S.: methodology, formal analysis, and investigation. M.S.: methodology, formal analysis, and investigation. All authors have read and agreed to the published version of the manuscript.

**Funding:** This work was carried out with the financial support of the Russian Science Foundation (grant no. 21-77-00028 and 19-17-00058) and programs of the POI FEB RAS (121021500052-9, AAAA-A20-120011090005-7).

**Institutional Review Board Statement:** Not applicable.

**Informed Consent Statement:** Not applicable.

**Data Availability Statement:** Any datasets not included in this published work are available upon reasonable request from the corresponding author; however, all data produced or analyzed during this investigation are contained in it.

**Conflicts of Interest:** The authors declare no conflict of interest.

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
