# Peer review of "Radium Isotopes and Hydrochemical Signatures of Surface Water-Groundwater Interaction in the Salt-Wedge Razdolnaya River Estuary (Sea of Japan) in the Ice-Covered Period"

_water, doi:10.3390/w15091792_

Round 1

Reviewer 1 Report

The article is well-written and is a good scientific piece of groundwater science applied research. The authors utilize the Ra isotopes and other physico chemical measurements to elucidate the surface water groundwater interaction in coastal basins. I recommend some minor changes, which may greatly enhance your article.  

In Figures 1 b and c you are missing the scale bar and North arrow, longitudes, and latitudes for Fig 1 c.

In the methodology, mention the model of the instrument(s) that have been used in the field and laboratory (pH, total alkalinity (TA), nutrients….etc.).

In line 104: Also, mention the detection limits for the measured nutrients (silica, phosphates, nitrates, nitrites, ammonium, Ptotal, and Ntota).

In Figure 2, indicate in the figure’s caption that station site locations are indicated in Figure 1.

The time indicated in the figure indicates 22 Jan 2022, 23 Feb 2022, ….etc.

In Figures 4 and 5, if possible, can you label the dots of surface and groundwater with the station number? This option is available in Excel. If not, you can use an excel chart labeler.

Figures 4, 5, and 6 are very nice figures; however still recommend labeling the plotted dots with station numbers.

In Line 279, please cite this article:

Groundwater recharge and salinization in the arid coastal plain aquifer of the Wadi Watir delta, Sinai, Egypt/ Applied Geochemistry 71 (2016).

The conclusion section needs more elaboration to reflect the excellent finding results of your research.  

Reviewer 2 Report

This is an interesting manuscript in which the authors try to elucidate the effects of groundwater discharge in an estuary. 

Overall, the paper is interesting because they use the winter period, when part of the river is frozen, to measure the effects of this discharge. Results shown in the text apparently demonstrate the groundwater discharge together with several ions and nutrients. However, they do not try to evaluate the percentage of groundwater in the total volume of the water river. Moreover, I recommend the authors to highlight the novelty of this manuscript in the results. 
